# Relationship between the Electrical Characteristics of Molecules and Fast Streamers in Ester Insulation Oil

**DOI:** 10.3390/ijms21030974

**Published:** 2020-02-01

**Authors:** Kaizheng Wang, Feipeng Wang, Ziyi Lou, Qiuhuang Han, Qi Zhao, Kelin Hu, Zhengyong Huang, Jian Li

**Affiliations:** State Key Laboratory of Power Transmission Equipment & System Security and New Technology, School of Electrical Engineering, Chongqing University, Chongqing 400044, China; 20131002008@cqu.edu.cn (K.W.); louziyi@cqu.edu.cn (Z.L.); hanqiuhuang@cqu.edu.cn (Q.H.); zhaoqi@cqu.edu.cn (Q.Z.); 20140802026@cqu.edu.cn (K.H.); huangzhengyong@cqu.edu.cn (Z.H.); lijian@cqu.edu.cn (J.L.)

**Keywords:** ionization potential, electron affinity, streamer, lighting impulse breakdown, ester insulation oil

## Abstract

The effects of C=C, ester and β-H groups on the ionization potential (*IP*) and electron affinity (*EA*) of molecules in natural ester insulation oil were investigated by density functional theory (DFT). The major contribution to the highest occupied molecular orbital (HOMO) comes from the carbon atoms adjacent to C=C. Thus, the *IP*s of triglycerides decrease as the number of C=C double bonds increases. The C=C in alkanes may also lower the *IP*. However, the β-H in triglycerides has little effect on the *IP*, and C=C and β-H have only a small effect on the *EA*s of the triglycerides because of the major contributions of atoms near the ester group in triglycerides to the lowest unoccupied molecular orbital (LUMO). This study calculated the *IP*s of 53 kinds of molecules in FR3, which are significantly lower compared with those of molecules in mineral oil (MO) and trimethylolpropane triester without C=C. However, the lightning impulse breakdown voltage (LI *V*_b_) of trimethylolpropane triester is still significantly lower than that of MO at the large gap. Therefore, the transition from slow to fast streamers under low lighting impulse voltage is determined by the ester group rather than by C=C and β-H. The ester group may attract more electrons, impacting itself more compared to alkane in MO and facilitating the transition from slow to fast streamers.

## 1. Introduction

Several billion liters of insulation oils have been widely applied in transformers, resistors, capacitors and thyristors, etc. [1]. An insulation oil acts as both insulation protection and as a heat transfer medium in power equipment; thus, it determines the equipment’s life span. At present, mineral insulation oil (MO) is still widely used in transformers, which are highly important in safe power grid operation. However, many leaks and combustion accidents of the transformer have been attributed to the MO’s low flash point, poor degradation and non-renewability. To improve the safety and reduce the environmental impact of the transformer, alternative insulation oils are demanded [2]. As the promising substitutes for MO, natural ester insulation oil (NEO) has several advantages of renewability and biodegradability and less flammability compared with MO [3]. In addition, NEO is claimed to delay the aging rate of insulation paper and prolong the lifetime of a transformer [4,5,6,7,8,9].

To date, NEO has mainly been applied in medium-voltage transformers; however, there is an increasing interest in applying NEO in large power transformers. Transformer industries have already successfully applied NEO in 420 kV and 238 kV [10]. In face of this application, lightning impulse (LI) breakdown characteristics of NEO need to be further studied [11], which are essential parameters for insulation design.

The LI *V*_b_ of ester insulation oil is comparable to that of MO in a quasi-uniform field [12,13,14]. However, in a non-uniform field, the LI *V*_b_ of NEO is considerably lower than that of MO for large oil gaps [11,15,16], which limits the application of NEO in large power transformers. Therefore, the underlying mechanisms of this phenomenon have been receiving increasing attention. The comparison of pre-breakdown phenomenon (streamers) characteristics in MO and NEO has been reported [15,17,18,19]. The inception voltage of streamers in NEO is lower under both polarities compared with that of MO in the non-uniform field. Streamer propagation in NEO evolves easily to fast mode, which results in its considerable low LI *V*_b_ at large oil gaps [16,20]. Comparisons of discharge parameters between NEO and MO were performed by many researchers. However, the reasons why streamer propagation in NEO evolves more easily to fast mode compared with that in MO is still unknown. Therefore, it is necessary to study the relationship between the electrical characteristics of the molecules and fast streamers in insulation oils. 

The *IP* and *EA* of molecules are related to the propagation of streamers in insulation oils [21,22]. Thus, density functional theory (DFT) was applied to calculate the *IP* and *EA* values of the molecules; DFT is the most widely used theoretical method to simulate the properties of both organic and inorganic molecules [23]. Specifically, the C=C, ester and β-H groups are the important groups in NEO. Therefore, in this work, the effects of the C=C, ester and β-H groups on the *IP* and *EA* values of molecules in NEO were investigated. Then, comparisons of the LI *V*_b_ and the *IP* distribution among the synthetic ester, natural ester (FR3) and MO were conducted to discuss the influence of molecular structure on the LI *V*_b_. 

## 2. Results

### 2.1. Effect of C=C on the Ionization Potential and Electron Affinity

As shown in Figure 1a, the *IP*s decrease as the number of C=C increases. The *IP* and *EA* of ten kinds of triglycerides were calculated: N-SSS, N-SSO, N-SOO, N-OOO, N-SSL, N-SLL, N-LLL, N-SSLn, N-SLnLn and N-LnLnLn. The rate of decline for the *IP*s of the triglycerides that included monounsaturated fatty acids is slightly higher than that of the triglycerides that included polyunsaturated fatty acids. The difference in *IP* between the N-SSS and N-LnLnLn is 1.12 eV. The decrease in triglyceride *IP* caused by an increase in *DN* is not linear. Initially, the *IP* decreases almost linearly, but the decline in IP tends to be flat as the number of C=C increase. Figure 1b shows the effect of C=C on the *EA* of triglycerides. The value of *EA* fluctuates as *DN* increases. The influence of C=C on the *EA* is small, as indicated by the small variation range of *EA* (0.16 eV).

### 2.2. Effect of β-hydrogen and Chain Length on the Ionization Potential and Electron Affinity

In triglycerides, the β-hydrogen atom (cf. Figure 2a) is present in the glycerol, which lowers the oxidation stability of the NEO. In the synthetic TME oil, trimethylolpropane is used to replace of glycerol in triglycerides. Thus, the synthesized TME oil is more stable than the NEO. Therefore, β-hydrogen is a crucial group for triglycerides in NEO, and it is necessary to study its influence on the *IP* and *EA*.

Although β-hydrogen has a significant effect on the oxidation stability of triglyceride molecules, its effect on the electrical properties of triglyceride molecules is unclear. The *IP*s and *EA*s of triglyceride molecules (N-CoCoCo, N-CyCyCy, N-CrCrCr, N-LaLaLa, N-MMM, N-PPP) and trimethylolpropane triester molecules (T-CoCoCo, T-CyCyCy, T-CrCrCr, T-LaLaLa, T-MMM and T-PPP) were calculated. Figure 3 shows the effects of β-hydrogen and the length of the carbon chain on the *IP* and *EA*. The number of carbons (*CN*) was used to represent the chain length of the molecule.

As shown in Figure 3a, β-hydrogen has little effect on *IP*. At the same *CN*, the triglycerides with β-hydrogen have a lower *IP* compared with that of trimethylolpropane triester (without β-hydrogen). The difference in *IP* between triglycerides and trimethylolpropane triester increases as the *CN* increases from 21 to 54. As the chain length increases, the *IP* of the molecule decreases. The *IP* decreases by approximately 7% as *CN* increases from 21 to 54. Therefore, the effect of chain length on the *IP* is significant. As shown in Figure 3b, the *EA* increases approximately linearly with increasing of the chain length. The *EA* of triglycerides with β-hydrogen is approximately 0.16 eV higher than that of trimethylolpropane triester without β-hydrogen.

### 2.3. HOMO and LUMO Characteristics

Koopman’s theorem can be used to roughly estimate the *IP* and *EA* of the molecule. According to Koopman’s theorem, the HOMO represents the molecule’s ability to donate an electron, and LUMO, as an electron acceptor, represents the molecule’s ability to obtain an electron; that is, for the isosurface of the HOMO of the molecule, the greater the contribution of the atom to the HOMO, the more likely it is that ionization will occur around it. Therefore, to clarify the influence of C=C and β-hydrogen on the *IP* and *EA*, it is necessary to analyze the HOMO and LUMO of the molecules (Figure 4). Stearic, oleic and linoleic fatty acids exist in N-OSLn. Therefore, analyzing the contribution of the atoms of N-OSLn (Figure 4a) can clearly clarify the phenomenon in which the rate of decline for the *IP*s of the triglycerides with monounsaturated fatty acids is slightly higher than that of triglycerides with polyunsaturated fatty acids (Figure 1a).

It was noticed that the *EA*s of N-OSLn and T-CyCyCy are negative, which indicates that their anions are unstable. As shown in Figure 4a, the HOMO is localized at the C=C. The Hirshfeld method [24] was used to calculate the HOMO and LUMO compositions. The major contribution to the HOMO comes from the carbon atoms (C100 and C102) adjacent to the C=C in the oleic acid side chain, and whose contributions are 33.45% and 33.43%, respectively. Therefore, the *IP*s decrease as the number of C=C increases (Figure 1a). The atoms in the linoleic acid side chain make no contribution to the HOMO; this fact can be used to clarify the phenomenon that the rate of decline for the *IP*s of the triglycerides that include monounsaturated fatty acids is slightly greater than that of the triglycerides that include polyunsaturated fatty acids (cf. Figure 1a).

As shown in Figure 4a, the atoms near the ester group in the linoleic acid side chain provide the main contribution to the LUMO. According to the Hirshfeld method, the contributions of H3 and H8, which near the ester group in the linoleic acid side chain, are 11.9% and 15.1%, respectively. It can be speculated that the *EA* of triglycerides is determined by the ester group. The atoms near the C=C make no contribution to the LUMO. Therefore, the influence of C=C on the *EA* is small, as shown in Figure 1b.

Although β-hydrogen significantly lowers the oxidation stability of NEOs, it has little effect on the *IP*, as shown in Figure 3a. Figure 5 shows the isosurface for the HOMO of N-CoCoCo. The atoms near the ester group from the position of *sn*-2 of N-CoCoCo without C=C are the major contributors to the HOMO. According to the Hirshfeld method, the major contributions to the HOMO come from C14, O15 and O9, whose contributions are 10.20%, 65.01% and 7.69%, respectively. The contribution of β-hydrogen (H6) is only 0.19%. Therefore, β-hydrogen has little effect of on *IP* (cf. Figure 3a). Likewise, the atoms near the ester groups at the positions of *sn*-1 and *sn*-3 of N-CoCoCo are the major contributors to the LUMO. H3 and H8 make major contributions to the LUMO (H3, 14.44%; H8, 14.42%), while the contribution of β-hydrogen (H6) to LUMO is only 0.55%. Therefore, β-hydrogen and chain length (*CN*) have little effect of on *EA* (cf. Figure 3b).

Figure 6 shows The HOMO energies of triglyceride molecules (N-CoCoCo, N-CyCyCy, N-CrCrCr, N-LaLaLa, N-MMM and N-PPP) and trimethylolpropane triester molecules (T-CoCoCo, T-CyCyCy, T-CrCrCr, T-LaLaLa, T-MMM and T-PPP). It is noted that the HOMO energies of the molecules are negative and increase slightly as *CN* increases, and the triglycerides have a higher HOMO energy compared with that of trimethylolpropane triester at the same *CN*. Koopman’s theorem holds that the *IP* of a molecule is equal to the opposite number of the HOMO orbital energy level of the molecule. The basic premise of Koopman’s theorem is that it assumes that the molecular orbital is not affected by the electrons escaping from the molecule; that is, it is assumes that the molecular orbital energy level does not change when the molecule loses an electron. However, in general, when a molecule gains or loses electrons, its HOMO energy is usually strongly affected. Therefore, the *IP*s obtained by Koopman’s theorem are usually inaccurate. However, Koopman’s theorem does reveal the negative correlation between the *IP* and the energy of the HOMO. Therefore, the *IP*s of molecules will increase slightly as *CN* increases, which clarifies why the *IP*s of the molecules decrease with an increase in *CN* and why the triglycerides have a lower *IP* compared with that of trimethylolpropane triester at the same *CN* (cf. Figure 3a).

### 2.4. Ionization Potential Distribution of Different Insulation Oils

Figure 7 shows the distribution of the *IP* for FR3. It is observed that the triglyceride molecules in FR3 are divided into three groups: triglycerides with three unsaturated acid chains—that is, the positions *sn*-1, 2 and 3 are all unsaturated fatty acids, such as N-LLL (three unsaturated chains (Three-U)); triglycerides with two unsaturated acid chains, where two of the *sn*-1, 2, and 3 positions are unsaturated fatty acids, such as N-SLL (Two-U); and triglycerides with one unsaturated acid chain in which one of the *sn*-1, 2 or 3 positions is an unsaturated fatty acid, such as N-SSL (One-U). The *IP* range for One-U triglycerides is 7.83 eV to 8.07 eV, at a concentration of 5.3%. The *IP* range for Two-U triglycerides is 7.37–7.73 eV, at a concentration of 39.5%, which is higher than that of One-U. The concentration for Three-U with a low *IP* (7.32–7.45 eV) is the highest (55.2%).

According to the above results, it was found that the atoms near C=C were the main contributors to the HOMO. This result indicates that C=C is critical in the *IP* of the triglyceride. To study the effect of C=C on the LI *V*_b_ of NEO, we prepared a new insulation oil (trimethylolpropane triester) without C=C (cf. Figure 8). The *IP*s of (T-CoCoCo, T-CyCyCy and T-CrCrCr) were calculated. As shown in Table 1, the *IP*s of the three molecules vary from 8.82 to 9.08 eV, which is significantly higher than that of the triglycerides in FR3.

Pyrene was used as the typical aromatic molecule at a concentration of approximately 5%. Hexadecane was used as the typical alkane molecule at a concentration of approximately 95%. As shown in Table 1, it can be noticed that the *IP* of the aromatic component accounting for 5% is between 7.14 eV and 7.73 eV, which is in the same range as that of Two-U and Three-U. According to our study, the C=C significantly lowers the *IP* of the triglyceride. The *IP* of benzene (C_6_H_6_) with three C=C is 9.24 eV [26], which is lower than that of cyclohexane (C_6_H_12_, 9.8 eV [27]). Therefore, the C=C in alkanes may also lower the *IP*. The numbers of C=C and carbons in pyrene are 8 and 16, respectively, while the numbers of C=C and carbons in N-OOO are 3 and 54, respectively. Therefore, we think the result shows that the *IP* of the aromatic component of MO that is in the same range of Two-U and Three-U is a coincidence.

The *IP* of the alkane component at a concentration of approximately 95% is about 9.08 eV to 9.15 eV. By comparing the *IP* distributions of the three insulation oils, it is found that the *IP*s of the triglycerides in FR3 are significantly lower than those of the alkane component (9.08–9.15 eV) and the trimethylolpropane triester (8.82 to 9.08 eV).

A standard LI voltage 1.2 (±30%)/50 (±20%) µs was applied in point-plane geometries. A tungsten needle electrode with a curvature radius of 100 ± 10 µm was used. The oil gap distance varied from 10 to 100 mm. Although the *IP* of TME oil is comparable to that of mineral oil, the LI *V*_b_ of TME oil is still significantly lower at large gaps (>50 mm) compared with that of MO (Figure 9). The LI *V*_b_ of ester insulation oils at a large gap of 100 mm is only half of that of mineral oil. However, there is little difference in LI *V*_b_ between FR3 and TME oil without C=C. Therefore, the C=C bond in NEO is not the main factor causing the low LI *V*_b_ the at the large gaps. Instead, the ester group in NEOs leads to the low LI *V*_b_ of NEO.

### 2.5. Formation Mechanism of Fast Streamers

According to [28], the streamer propagations are divided into 1st, 2nd, 3rd and 4th modes. The 1st and 2nd modes are slow streamers, but the fast streamers (the 3rd and 4th modes) are likely responsible for LI *V*_b_ at large gaps [28]. As shown in Figure 5, the LI *V*_b_ of ester insulation oil [22] at the large gaps (>50 mm) is significantly lower than that of MO. Therefore, it is necessary to clarify the formation mechanism of fast streamers in ester insulation oils (TME and natural ester oil).

To date, there is still no systematic research on the initial voltage (*V*_3_) of 3rd-mode streamers. When the applied voltage is greater than *V*_3_, the velocity of streamer propagation with a strong light emission increases suddenly [20]. The propagation field of the 3rd-mode streamers in NEO at the streamer tip is estimated to be 10–20 MV/cm [16,20], which is far from the field ionization of the insulation oils [29]. At 10–20 MV/cm, impact ionization can take place [30]. Impact ionization may promote the slow streamers to fast streamers.

At 10–20 MV/cm^−1^, the typical molecules of the insulation oil molecules are ionized by the impact of the electrons. Therefore, the *IP*s of molecules are less important than their *EA*s. Thus, it is necessary to compare the LUMO isosurface between the MO and ester oil. As shown in Figure 10, the distribution of atomic contributions to LUMO by hexadecane and pyrene is relatively uniform, which is different from that of triglycerides and trimethylolpropane triester. There are eight atoms in hexadecane whose contributions are greater than 5% (C2, C6, C7, C10, C11, C14, C16 and C17), which is about 31% of the total number of atoms. However, the numbers of atoms in pyrene and hexadecane whose contributions are greater than 0.5% are only two (H21 and H25) and one (H2), respectively. In summary, the major contribution to LUMO comes from the atoms near the ester group. The distribution of atomic contributions in hydrocarbon molecules (MO) is more uniform than that in triglycerides and trimethylolpropane triesters.

Because the major contribution to LUMO comes from the atoms near the ester group, the ester group in ester insulation oil is impacted by more electrons compared to that of the alkane in MO (Figure 11). At large gaps, impact ionization can play a major role in streamer development. For the ester insulation oil, the voltage needed to generate a large number of electrons is lower than that for MO. Therefore, more electrons and positive ions are generated in ester insulation oil, which leads to their relative ease in producing fast streamers.

## 3. Materials and Methods

### 3.1. Materials

Triglycerides are the main constituents of NEO. Figure 2a shows the molecular geometry of triglycerides with β-H, which significantly lowers the oxidation stability of NEO [3,31,32]. The D, D^’^ and D” in the molecule represent the fatty acids, which always contain C=C. The different stereochemical positions (*sn*-1, 2 or 3) on the glycerol backbone are the locations corresponding to the three fatty acids. N-DD’D” is used to represent the triglycerides, and the “N” stands for natural ester. For example, N-OOO corresponds to the glycerol tri-oleate for the NEO. Table 2 shows the abbreviations for different types of fatty acids.

The electrochemical properties of molecules in the insulation oil determine the characteristics of streamer propagation [33,34,35]. Therefore, it is necessary to investigate the distribution of *IP* in different insulation oils. NEO (FR3) is prepared from refined soybean oil, which is a mixture of approximately 53 kinds of triglycerides. The concentrations of 53 kinds of triglycerides in soybean oil were measured in [36]. To study the distribution of *IP* in FR3, the *IP*s of 53 kinds of triglycerides were calculated by DFT.

Trimethylolpropane triester (TME) was prepared by the authors group, which was synthesized by esterification of the saturated fatty acids and trimethylolpropane [25] (Figure 8). As shown in Figure 2b, the X, X^’^ and X” in the molecule represent the fatty acids without C=C. T-XX’X” represents the trimethylolpropane triester. For example, T-CyCyCy stands for trimethylolpropane tri-caprylate.

MO is a mixture of various hydrocarbon molecules. Mineral oils mainly consist of hydrocarbon compounds with different structures, such as alkanes (paraffinic and naphthenic hydrocarbons) and aromatics, as shown in Figure 2c. Pyrene was used as the typical aromatic molecule at a concentration of approximately 5% [37]. Hexadecane was used as a typical alkane molecule at a concentration of approximately 95%.

### 3.2. Methods

DFT has previously been successfully applied to calculate the properties of molecules [21,38]. The Gaussian 09 W software package [39] was used to perform quantum chemical calculations. The Becke, 3-parameter, Lee–Yang–Parr (B3LYP) functional method [40] with the 6-31+G* basis set was used to calculate the molecule properties. The *IP* of a molecule is defined as the energy required to cause a neutral molecule M to lose one electron and form a positive ion M^+^. Thus, the *IP* is expressed as follows:M→M^+^ + *e*; *IP* = *E*^M+^ − *E*^M^(1)

First, the ground-state geometries of M in the gas phase were optimized by B3LYP/6-31+G* to obtain the energy *E*^M^ of the neutral molecule. Second, the optimized molecule was used to calculate the energy *E*^M+^. Finally, the *IP* of the M molecule was calculated according to Equation (1), and the *EA* of the molecule was calculated according to Equation (2).*EA* = *E*^M^ − *E*^M−^(2)
where *E*^M^ is the energy of the M molecule in the ground-state, and *E*^M−^ is the energy of the anion M^−1^ as calculated by B3LYP/6-31+G*.

No prior experimental data existed related to the *IP* and *EA* values of triglycerides, trimethylolpropane triester and hexadecane. Therefore, to validate the DFT method, a calculation of vibration frequencies for the functional groups in N-OOO was performed. The relative error between the calculated and measured values [41] is approximately 2% (Table 3). Furthermore, the calculated *IP* of pyrene is 7.14 eV, which is only about 0.28 eV lower than that of the experimental value (7.42 eV [42]). Thus, B3LYP/6-31+G* was adopted to conduct the further calculations.

## 4. Conclusions

In this work, the effects of the C=C, ester and β-H groups on the *IP* and *EA* of molecules in NEOs was investigated. Comparisons of LI *V*_b_ and the *IP* distribution of synthetic esters, natural esters (FR3) and MO oils were conducted. The major contribution to the highest occupied molecular orbital (HOMO) comes from the carbon atoms adjacent to C=C. Thus, the *IP*s of triglycerides decrease as the number of C=C double bonds increases. The C=C in alkanes may also lower the *IP*.The influences of C=C and β-H on the *EA* of the triglyceride is small, as indicated by the small variations in *EA* (0.16 eV). This occurs because the contributions of β-H and the atoms near C=C to the LUMO are small. Instead, the atoms near the ester group make a primary contribution to the LUMO.β-H has little effect on the *IP* of the triglyceride because it makes no contribution to the HOMO. Under the same chain length, the maximum difference between the triglyceride (with β-H) and trimethylolpropane triesters (without β-H) is only 0.6%. As the molecular chain length increases, the ionization potential decreases. The *IP*s of triglycerides are determined by the C=C and chain length.In this study, the *IP*s of 53 kinds of molecules in FR3 were calculated. The *IP*s of molecules of the trimethylolpropane triester molecules are significantly higher compared with those of molecules of FR3. However, the LI *V*_b_ of the trimethylolpropane triester is still significantly lower than that of MO at large gaps. Therefore, neither the C=C or β-H but rather the ester group leads to this phenomenon. The atoms near the ester group make the main contribution to the LUMO, which draws more electron impact toward themselves compared to the alkane in MO. This characteristic promotes the transition from slow to fast streamers.

## Figures and Tables

**Figure 1 ijms-21-00974-f001:**
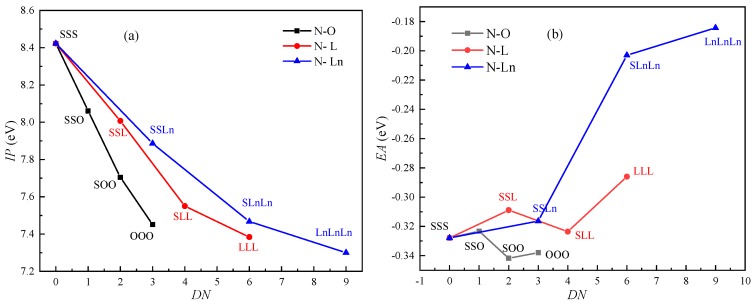
Effect of C=C on the ionization potential (*IP*) and electron affinity (*EA*) of triglycerides (*DN*: the number of C=C): (**a**) *IP*; (**b**) *EA*.

**Figure 2 ijms-21-00974-f002:**
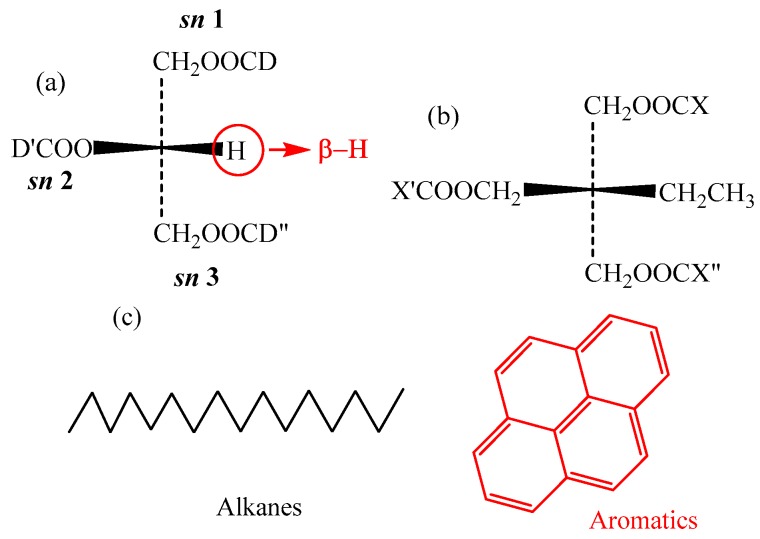
Molecular geometry of different insulation oil: (**a**) Natural ester insulation oil; (**b**) trimethylolpropane triester; (**c**) mineral oil.

**Figure 3 ijms-21-00974-f003:**
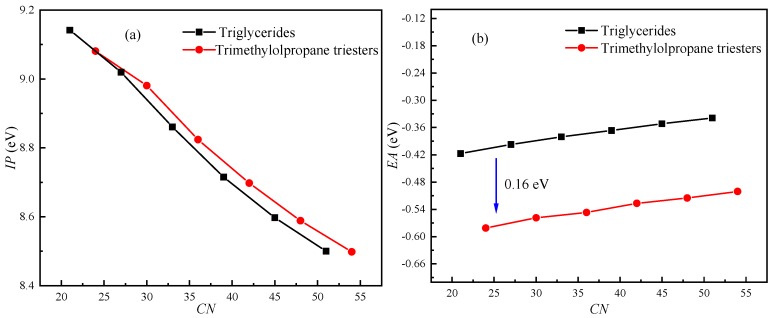
Effects of β-hydrogen and chain length (*CN*) on *IP* and *EA*: (**a**) *IP*; (**b**) *EA*.

**Figure 4 ijms-21-00974-f004:**
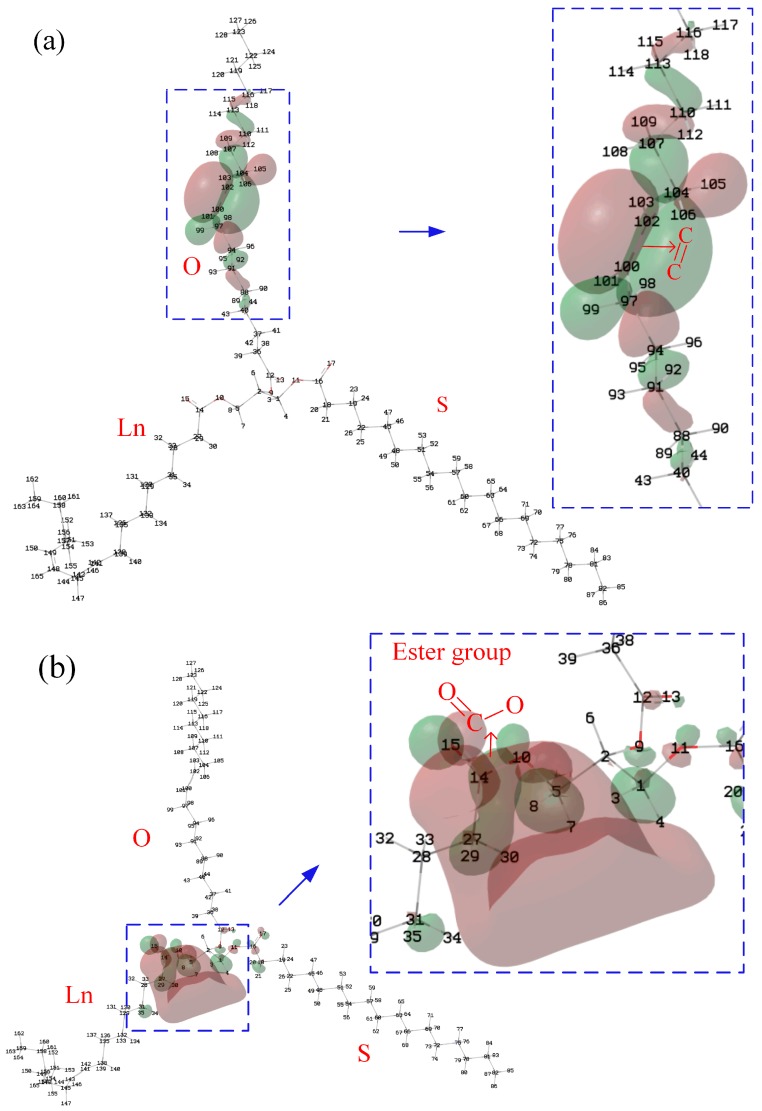
Isosurface of the HOMO and LUMO of N-OSLn: (**a**) HOMO; (**b**) LUMO.

**Figure 5 ijms-21-00974-f005:**
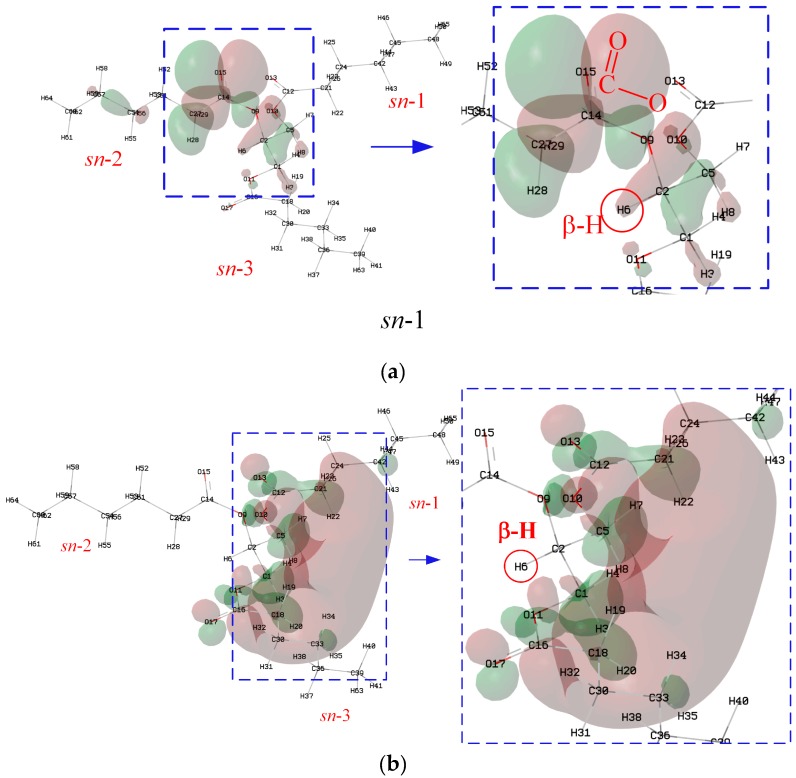
Isosurface of the HOMO and LUMO of N-CoCoCo: (**a**) HOMO; (**b**) LUMO.

**Figure 6 ijms-21-00974-f006:**
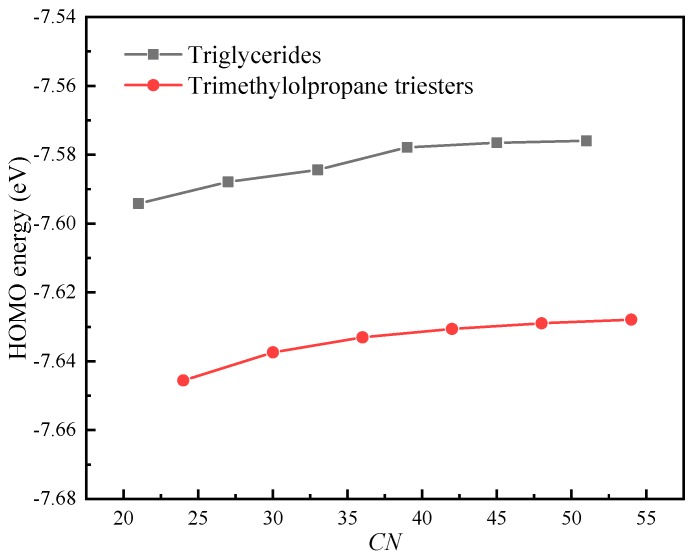
The HOMO energies of molecules.

**Figure 7 ijms-21-00974-f007:**
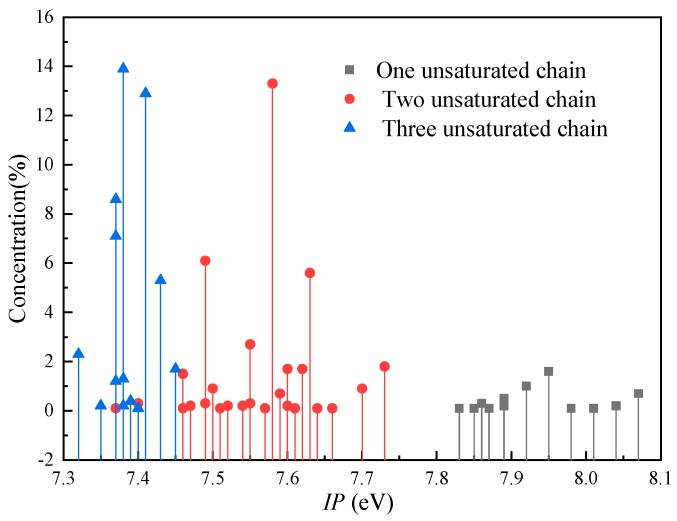
The distribution of the *IP* of the natural ester oil.

**Figure 8 ijms-21-00974-f008:**
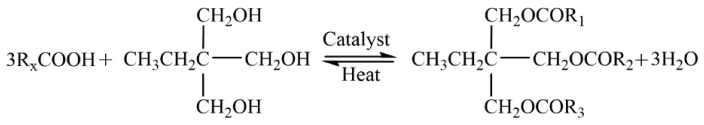
Reaction between the trimethylolpropane and saturated fatty acids [25].

**Figure 9 ijms-21-00974-f009:**
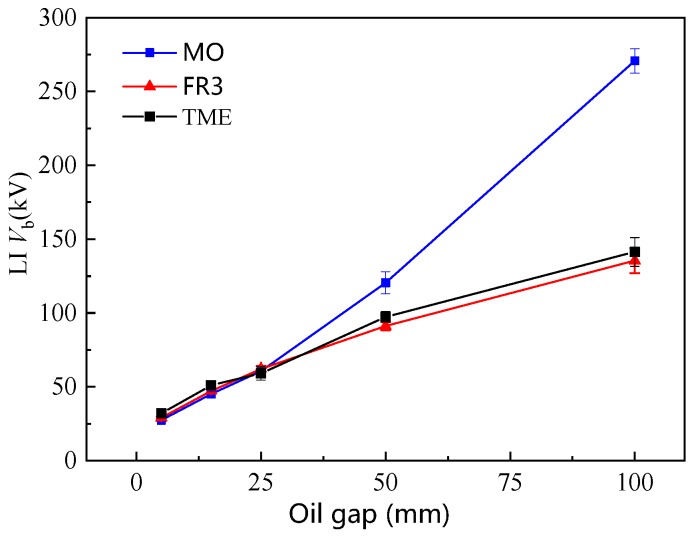
The impulse breakdown voltage at different oil gaps.

**Figure 10 ijms-21-00974-f010:**
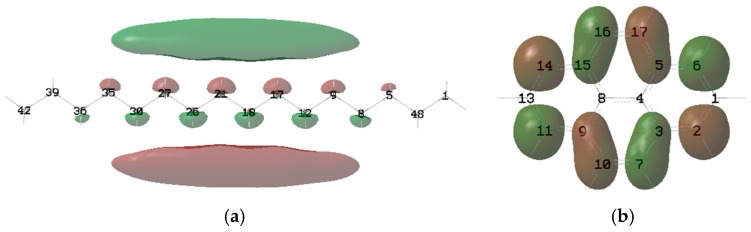
Isosurface of the LUMO of typical molecules of MO: (**a**) Octane; (**b**) pyrene.

**Figure 11 ijms-21-00974-f011:**
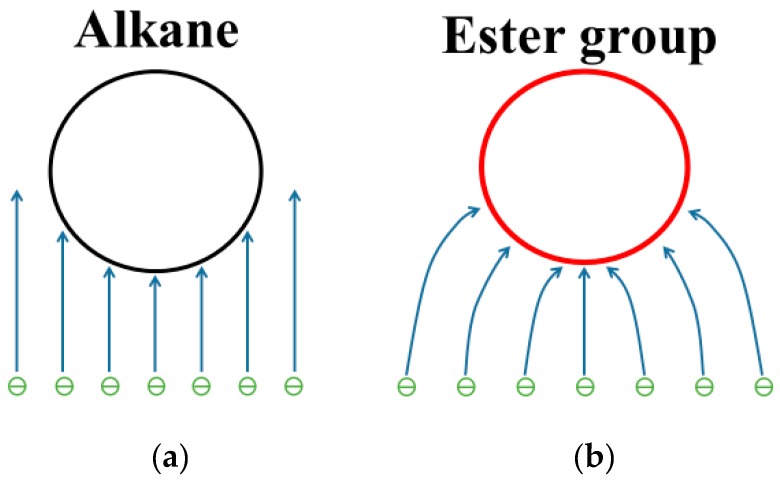
Sketch of the impact ionization in insulation oil: (**a**) MO; (**b**) ester insulation oil.

**Table 1 ijms-21-00974-t001:** The distribution of the *IP* of the three insulation oils.

*IP* (eV)	FR3	MO	TME
7.14–7.73	94.7%	5%	0
7.83–8.07	5.3%	0	0
8.82–9.08	0	0	100%
9.08–9.15	0	95%	0

**Table 2 ijms-21-00974-t002:** Abbreviations for different types of fatty acids.

Fatty Acid	Symbol	*CN*:*DN*^1^
Caproic	Co	6:0
Caprylic	Cy	8:0
Capric	Cr	10:0
Lauric	La	12:0
Myristic	M	14:0
Palmitic	P	16:0
Stearic	S	18:0
Oleic	O	18:1
Linoleic	L	18:2
Linolenic	Ln	18:3

^1^*CN*:*DN* = the number of carbon (CN):the number of C=C (DN).

**Table 3 ijms-21-00974-t003:** Vibration frequencies of C=O and C=C in triolein.

Methods	Vibrational Frequency/cm^−1^					
	C=O			C=C		
	*sn*-1	*sn*-2	*sn*-3	*sn*-1	*sn*-2	*sn*-3
B3LYP/6-31+G*	1747.3	1750.5	1757.4	1656.0	1656.3	1656.4
Experimental values	1746 [41], 1745 [43]			1653 [41], 1654 [43]		

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
