# Peer review of "Relationship between the Electrical Characteristics of Molecules and Fast Streamers in Ester Insulation Oil"

_ijms, 2020, doi:10.3390/ijms21030974_

Round 1

Reviewer 1 Report

My previous comments were addressed and introduced into the text. Also comments from other reviewers allowed for improving the paper. I believe that it is ready for publication.

Author Response

To: Editors and reviewers

Dear referees, dear editors,

Thank you very much for the excellent and professional revision of our manuscript. We sincerely appreciate all the reviewers and editor for taking the time and efforts to make this research more valuable. We seriously have taken all comments into consideration during preparing the point-to-point reply. The changes made and newly revision in the revised manuscript are marked as red for easy inspection. Detailed responses to editors and reviewers’ comments and advice are listed as follows.

Again, thank you for giving us the opportunity to strengthen our manuscript with your valuable comments and queries.

Best regards,

< Kaizheng Wang, Feipeng Wang, Qi Zhao, Ziyi Lou, Qiuhuang Han, Kelin Hu, Zhengyong Huang and Jian Li>

Reviewer 2 Report

I really appreciate the effort spent by the authors on revising their paper.

In particular, the information added in the results and discussion section significantly improved the readability of the manuscript and the understanding of the obtained results, which are now introduced and discussed in a more accurate and logical way.

Engligh has sensibly improved, and most of the errors and inaccuracies have been corrected. But there are still some little changes to make before publication:

1) Replace "Ionization Energy" with "Ionization Potential", coherently with the use of the acronym "IP".

2) Use "Electron Affinity" (and stop) instead of "Electron Affinity energy".

3) Define the acronym DFT in the abstract.

4) Try to improve the readability of the introductory section, for example you write "There is an increasing interest in applying NEOs in large power transformers. Transformer industries have successfully applied NEOs in large power transformers (420kV and 238 kV) [10]. In the face of the application in large power transformers, lightning impulse (LI) breakdown characteristics of NEOs need to further study [11]. LI breakdown voltage (Vb) of NEOs is an essential parameter of insulation design for large power transformers". As you can see, you repeat four times "large power transformers" in only five lines!

5) In describing molecules, at the beginning of section 2.1, you forgot to define what "N" stands for in N-DD'D".

6) At point 4 of the concluding remarks you mentioned "58 kinds of molecules" instead of 53. Where is the truth? 58 or 53?

7) Finally, I recommend to triple check the English before re-submission, because there are still some errors in the text (especially in the abstract, in the introductory section, and in the materials and methods section), as well as some sentences with no apparent meaning. Some examples:

"C=C and ß-H have only a little effect on the EA of the triglyceride because the major contribution of atoms near the ester group in triglycerides to the lowest unoccupied molecular orbital (LUMO)"? Probably "of" is missing after "because"?

"Trimethylolpropane triester (TME) prepared by the authors group, which was prepared by esterification of saturated fatty acids and trimethylolpropane [30]". It's a truncated sentence.

"wildly"? Maybe "widely".

"non-renewability" instead of "non-renewable".

"streamers" instead of "steamers".

"prolonge" instead of "prolong".

DFT "was been applied(!!!)".

Replace "anionic" with "anion" or with "negative ion".

Sometime "Koopman" appears as "Kupman", please check. ...

Author Response

This manuscript is a resubmission of an earlier submission. The following is a list of the peer review reports and author responses from that submission.

Round 1

Reviewer 1 Report

The results of LI withstand voltage in inhomogeneous field is correct and similar to our results.

TME is not defined.

3.4 Capital F, when "W"

Reviewer 2 Report

In this manuscript, Kaizheng Wang et al. introduce the results of DFT calculations of ionization potential (IP) and electron affinity (EA) of natural ester insulation oils, focusing on the influence on the calculated values of IP and EA of some peculiar features of the investigated molecules, such as the C=C bond and the ester group.

It’s an interesting study, mostly for the implication it could have on the possible exploitation of ester oils in high power transformers. However, the paper, at least in the submitted version, does not meet the quality standards for publication in IJMS, for the following reasons:

English used in some sections is very poor. It’s very hard to understand what authors want to mean, in some points. Some sentences are truncated, with no apparent meaning. The word “molecular” is very often used instead of “molecule”, this is not acceptable. But this only an example. I strongly recommend to improve the English used, because both language and style require extensive editing before possible consideration for publication. There are plenty of errors and inaccuracies in the organization of the paper. It seems it hasn’t even been checked before submission. Figure 2 is sometimes indicated as “Figure 1”, for example when describing the structure of mineral oils in page 3. There are two “Figure 2” (in page 3 and page 4), it’s a real mess. Triester molecules are sometimes indicated as “TM-“, some other times as “T-“. Titles of sections 3.1 and 3.2 should contain “Electron Affinity”, and not simply “Affinity”. In a few word, I would carefully check the paper before submission, because it’s not accurate enough in the present version. Results and their interpretation seem to be untied to each other. Just an example: data of figures 2 and 3 of page 4 (IP and EA as a function of DN and CN) are firstly introduced from a phenomenological point of view. Then, Figure 4 is discussed (in a bad way indeed, some sentences are truncated and not understandable), with no apparent logical connection to the results shown in fig. 2 and 3. In this case it should be clearly stated, for sake of clarity, that DFT calculations of the isosurfaces of HOMO and LUMO were performed to give a chemical-physical interpretation of the results shown in fig. 2 and 3. In the present version, it’s quite hard to notice a logical thread. I suggest to revise the paper paying more attention to give a more fluid and homogeneous presentation of the results on IP and EA and their connection to the molecular structures. More generally and frankly speaking, it’s as if the different sections were written by different authors without a final “harmonization” of the manuscript, which conversely is mandatory for improving its readability.

Minor comments:

Define the acronym “LI” in the abstract. Sometimes the term “proportion” is used to replace “concentration”. It’s not correct. What is a “types molecule”? Probably a “typical molecule”? Both IP and EA should be introduced in terms of energy, by writing that IP is the energy difference between HOMO and vacuum level, whereas EA is the energy difference between LUMO and vacuum level. At the beginning of section 3.2 (fifth line) it’s written “molecular affinity”. It’s wrong, it’s “electron affinity”. I do not understand why a 7% variation is considered “small” (i.e. the effect of chain length on IP), whereas a 13% variation, as the influence of DN on IP is considered significant. In my opinion, both are significant. By varying CN, IP decreases from 9.1 to 8.5 eV: it’s definitely not a small decrease. Why have N-OSL been used for isosurface calculations reported in Fig.4? Wouldn’t it be more consistent to use one of the molecules whose IP and EA values are reported in Figs. 2 or 3 (as in the case of T-CyCyCy)? Figure 4 should be enlarged. The first paragraph of Section 3.3 should be moved to introduction or to the section describing the methodology used. I would clearly write that a further demonstration of the influence of C=C bond on the IP value is given by the IP of the aromatic component of MO, which is in the same range of Two-U and Three-U FR3. This should be highlighted by the authors, in my opinion. What do the authors mean with “relatively average” when commenting on the distribution of atom contribution to LUMO in hexadecane and pyrene? Probably “relatively uniform”? Check the caption of Fig.8. The term “Figure 8” is missing.

Reviewer 3 Report

The paper presents the reseacr results on electrical properties of insulating liquids, especially under lightning impulse conditions. The topic is very interesting, as new insulating liquids are being introduced into technical insulation systems, which are especially important in power transformers, as one unit may contain almost 100 tons of oil (e.g. 450 MVA autotransformer contains 86 tons os oil). The application of a better insulating liquid, that would give better parametrs with cellulose (or Nomex) is still a challenge for the future. The results presented in the paper show avery important aspect of this problem.

In general paper is very interesting and the results are clear and well discussed. There, however, are some issues that can be improved:

English language can be corrected. I am not a native speaker in English, but in many places I can find grammatical errors ('was' instead of 'were', etc.). There are some editorial errors, the paper should be thoroughly checked, e.g. LI in abstract is not descibed, point 3.4. starts with small letter and others. I suggest adding into Introduction a brief background of liquid insulation application. A reader should now why the problem described in the paper is important from practical point of view and what are high voltage applications of described insulating liquids. Why FR3 or TME are being introduced into high voltage insulation, to replace mineral oils?